Regulation of plant gene expression by tsRNAs in response to abiotic stress

Li Chunmei 1 2 3
Zhu Jing 2 3
Jin Han 1 4
Feng Haotian 1 4
Zhuang Haimin 2 3
Du Zijun 2 3
Zhu Guolin 1 4
He Haiyang 2 3
Ye Fuyang 2 3
Mo Zhaohui 1 3 4
Hu Qingtao 1 4
Chen Zhenbang 2 3
Liu Kai 1 3 4 liukai5088@126.com
http://orcid.org/0000-0003-0892-8906 Wan Xiaorong 1 2 3 biowxr@126.com
1 Key Laboratory of Green Prevention and Control on Fruits and Vegetables in South China, Ministry of Agriculture and Rural Affairs, Zhongkai University of Agriculture and Engineering , Guangzhou , China
2 Guangzhou Key Laboratory for Research and Development of Crop Germplasm Resources, Zhongkai University of Agriculture and Engineering , Guangzhou , China
3 College of Agriculture and Biology, Zhongkai University of Agriculture and Engineering , Guangzhou , China
4 Innovative Institute for Plant Health, Zhongkai University of Agriculture and Engineering , Guangzhou , China
Nunes-da-Fonseca Rodrigo
Electronic publication date: 2025 May 23
Publication date: 2025
Volume: 13
Electronic Location ID: e19487
Received 2024 Dec 24; Accepted 2025 Apr 27
Copyright: © 2025 Li et al.
Copyright year: 2025
Copyright holder: Li et al.
License: This is an open access article distributed under the terms of the Creative Commons Attribution License, which permits unrestricted use, distribution, reproduction and adaptation in any medium and for any purpose provided that it is properly attributed. For attribution, the original author(s), title, publication source (PeerJ) and either DOI or URL of the article must be cited.
License URL: https://creativecommons.org/licenses/by/4.0/

Keywords: tRNA, tRNA-derived small RNAs (tsRNAs), Gene expression, Abiotic stress, Plant

Funding: National Natural Science Foundation of China 32402397, 32071737 and 32111530289 Basic and Applied Basic Research Foundation of Guangdong Province 2024A1515013028 and 2025A1515012591 Guangdong Provincial Universities Characteristic Innovation Project 2023KTSCX046 Guangzhou Science and Technology Plan Project 2024A04J4995 State Key Laboratory of Crop Gene Exploration and Utilization in Southwest China SKL-KF202315 State Key Laboratory for Managing Biotic and Chemical Treats to the Quality and Safety of Agro-products 2021DG700024-KF202408 Department of Science and Technology of Guangdong Province 2023B0202010025 Department of Education of Guangdong Province 2020ZDZX1013 Guangdong University Key Laboratory for Sustainable Control of Fruit and Vegetable Diseases and Pests KA21031C502 This work was supported by the National Natural Science Foundation of China [32402397, 32071737 and 32111530289], the Basic and Applied Basic Research Foundation of Guangdong Province [2024A1515013028 and 2025A1515012591], the Guangdong Provincial Universities Characteristic Innovation Project [No. 2023KTSCX046], the Guangzhou Science and Technology Plan Project [2024A04J4995], the State Key Laboratory of Crop Gene Exploration and Utilization in Southwest China [SKL-KF202315], the State Key Laboratory for Managing Biotic and Chemical Treats to the Quality and Safety of Agro-products [2021DG700024-KF202408], the Department of Science and Technology of Guangdong Province [2023B0202010025] and the Department of Education of Guangdong Province [2020ZDZX1013], and the Guangdong University Key Laboratory for Sustainable Control of Fruit and Vegetable Diseases and Pests [KA21031C502]. The funders had no role in study design, data collection and analysis, decision to publish, or preparation of the manuscript.

==============================
Objective

Transfer RNA-derived small RNAs (tsRNAs) are emerging regulators of gene expression in response to abiotic stress. This review aims to summarize recent advances in the classification, biogenesis, and biological functions of tsRNAs, with a focus on their roles in plant stress responses and the methodologies for investigating these molecules.

Methods

We conducted a comprehensive literature search across PubMed, Web of Science, and Google Scholar using keywords such as “tRNA-derived small RNAs”, “abiotic stress”, “plant gene regulation”, and “RNA sequencing”. Studies were selected based on their relevance to tsRNA biogenesis pathways, stress-responsive mechanisms, and functional validation in plant systems. Classification of tsRNAs was performed according to cleavage site specificity and nucleotide length. Bioinformatic tools and experimental approaches for tsRNA identification, target prediction, and functional validation were evaluated.

Results

tsRNAs are categorized into two main types: tRNA-derived stress-induced RNAs (tiRNAs; 29–50 nt) and tRNA-derived fragments (tRFs; 14–40 nt). tiRNAs arise from anticodon loop cleavage by RNase A/T2, while tRFs are generated via Dicer-dependent or -independent pathways. These molecules regulate gene expression at transcriptional, post-transcriptional, and translational levels by interacting with AGO proteins, displacing translation initiation factors, and modulating stress granule assembly. In plants, tsRNAs respond dynamically to abiotic stresses (e.g., drought, salinity, heat), influencing stress signaling pathways and epigenetic modifications. Advanced sequencing techniques (e.g., cP-RNA-seq, RtcB sRNA-seq) and databases (PtRFdb, tRFanalyzer) have facilitated tsRNA discovery and functional annotation.

Conclusions

tsRNAs represent a versatile class of regulatory molecules in plant stress biology. Their ability to fine-tune gene expression underpins adaptive responses to environmental challenges. Future research should prioritize standardized methodologies for tsRNA profiling, elucidation of stress-specific biogenesis mechanisms, and exploration of their potential as biomarkers or therapeutic targets for crop improvement. Integrating tsRNA research with systems biology approaches will deepen our understanding of plant resilience mechanisms.

Introduction

Transfer RNA (tRNA) is an abundant type of small RNA, accounting for 4–10% of all cellular RNAs (Kirchner & Ignatova, 2015; Shi et al., 2023). tRNA biogenesis initiates with RNA polymerase III transcribing tRNA loci, producing precursor tRNAs (pre-tRNAs) that include 5′ leader and 3′ trailer sequences. The 5′ leader and 3′ trailer sequences are cleaved by RNase P and RNase Z, respectively (Frank & Pace, 1998; Schramm & Hernandez, 2002; Ceballos & Vioque, 2007; Liu et al., 2021a). Subsequently, intron sequences are excised from pre-tRNAs, and tRNA nucleotidyltransferase appends a ‘CCA’ sequence to the 3′ end of the tRNA (Weiner, 2004; Abelson, Trotta & Li, 1998). Finally, after various post-transcriptional modifications, mature tRNAs are formed. These modifications include methylation, where a methyl group is added to specific nucleotides in the tRNA, affecting its structure and stability, as well as its interaction with other cellular components (Lorenz, Lünse & Mörl, 2017; Suzuki, 2021). Other modifications include isopentenyl adenylation, the addition of an isopentenyl group to adenine residues, and thiouridylation, the conversion of uridine to thiouridine (Frye et al., 2018; Ohira et al., 2022; Yared, Marcelot & Barraud, 2024). These modifications are crucial for the proper functioning of tRNAs in translation and other cellular processes. Mature tRNAs comprise a displacement loop (D-loop), T-loop, anticodon loop, and variable loop (Phizicky & Hopper, 2010). Plants respond to environmental stresses through a complex network of signaling pathways and physiological changes. For example, under drought stress, plants may activate the abscisic acid (ABA) signaling pathway, which leads to the closure of stomata to reduce water loss and the induction of genes involved in drought tolerance (Zhu, 2002; Park et al., 2019). Similarly, under salinity stress, plants may activate ion homeostasis mechanisms to maintain cellular osmotic balance (Xu et al., 2015). These signaling pathways often involve the production of reactive oxygen species (ROS) and the activation of antioxidant defense systems to protect cellular components from damage (Basit et al., 2024; Pandey & Zhu, 2024). MicroRNAs (miRNAs) are important non-coding RNAs that participate in the regulation of plant gene expression under stress conditions (Sunkar & Zhu, 2004; Matsui et al., 2013). For example, miR168 targets the transcripts of ARGONAUTE1 (AGO1), which is involved in RNA silencing, and miR398 targets the transcripts of copper/zinc superoxide dismutase (Cu/Zn SOD), which is involved in ROS scavenging (Iki et al., 2018; Li et al., 2019; Miao et al., 2022; Zhou et al., 2022; Zhao et al., 2023). These miRNAs help plants adapt to stress conditions by fine-tuning the expression of stress-related genes. Recent studies indicate that tRNAs generate tRNA-derived small RNAs (tsRNAs), which influence cellular mechanisms similarly to miRNAs which typically function by binding to complementary sequences on target mRNAs, leading to mRNA cleavage or translational repression. They are involved in various cellular processes, including development, cell proliferation, and stress response. Similarly, tsRNAs can also bind to target mRNAs and regulate their expression, thereby influencing cellular functions. This similarity in function suggests that tsRNAs may play a role in gene regulation similar to that of miRNAs, especially under stress conditions, such as drought, salt, high temperature stress. (Raina & Ibba, 2014; Zhu, Ow & Dong, 2018; Zhu et al., 2018; Guan & Grigoriev, 2020; Alves & Nogueira, 2021; Ma, Liu & Cao, 2021; Zhang et al., 2022; Cai et al., 2025; Pang et al., 2025).

tsRNAs are produced through the cleavage of a tRNA precursor or mature tRNA by specific ribonucleases, with the cleavage product typically ranging in length from 14 to 40 nucleotides (nt) (Zhu et al., 2018). Similar to miRNAs, which are well-known regulators of gene expression, and PIWI-interacting RNAs (piRNAs), which are small non-coding RNAs that interact with PIWI proteins (members of the Argonaute protein family) and are primarily involved in the defense against transposable elements and the maintenance of genome stability, tsRNAs can also be identified through small RNA sequencing (Keam et al., 2014; Ma et al., 2021a). Recent research indicates that tsRNAs are prevalent in both prokaryotes and eukaryotes, showing distinct spatiotemporal accumulation patterns essential for their functions. Their dysregulation can disrupt homeostasis (Kumar, Kuscu & Dutta, 2016; Zhu, Ow & Dong, 2018; Zhu et al., 2018; Tan, Tan & Duan, 2019; Su et al., 2020; Liu et al., 2021b; Ma, Liu & Cao, 2021; Ma et al., 2021b; Wang et al., 2023; Panstruga & Spanu, 2024). Investigating the biological roles and molecular mechanisms of tsRNAs has emerged as a key area of research in small non-coding RNAs. In this review, we explore recent advances in plant tsRNAs, with a focus on how they regulate gene expression in response to abiotic stress, and discuss remaining areas of exploration for tsRNA research.

Intended audience and need for this review

Despite the significant advancements in our understanding of plant stress responses, the role of tsRNAs (tiny subclass of small RNAs) in regulating gene expression under abiotic stress conditions remains an understudied area. tsRNAs have emerged as novel regulators of gene expression with potential implications in stress tolerance and crop improvement. However, current knowledge on their biogenesis, modes of action, and specific roles in stress response pathways is scattered and incomplete. This review consolidates recent findings and highlights the gaps in our understanding, thereby necessitating a comprehensive evaluation of the current research landscape. By providing an in-depth analysis of tsRNA-mediated gene regulation in plant stress responses, this review is crucial for those who are interested in understanding the intricate mechanisms by which plants regulate their gene expression in response to environmental stressors such as drought, salinity, temperature extremes, and other abiotic stresses.

Survey methodology

The authors conducted an exhaustive literature search across multiple databases including PubMed, Web of Science, and Google Scholar to gather relevant information on the regulation of plant gene expression by tRNA-derived small RNAs (tsRNAs) in response to abiotic stress. The search strategy incorporated both controlled vocabulary (subject headings) and free-text terms to ensure comprehensiveness. Key search terms utilized in this process were: “tRNA”, “tRNA-derived small RNAs (tsRNAs)”, “Gene expression”, “Abiotic stress”, “Plant”, “stress response mechanisms”, “transcriptional regulation”, “post-transcriptional regulation”, “small non-coding RNAs”, and “RNA processing”. Articles were screened by their titles, abstracts, and full texts where available, to identify studies that specifically addressed the role of tsRNAs in modulating gene expression under various abiotic stress conditions. This review is compiled from a meticulously curated selection of published works, which have been classified, organized, and critically analyzed based on their relevance to the study of tsRNA-mediated gene regulation in plants exposed to abiotic stresses.

Classification and biogenesis of tsRNAs

tsRNAs can be classified into two main types, tRNA-derived stress-induced RNAs (tiRNAs) and tRNA-derived fragments (tRFs), based on their length and the position of the cleavage site in their precursors (Table 1) (Lee et al., 2009; Telonis et al., 2015; Telonis et al., 2019; Magee & Rigoutsos, 2020; Pan, Han & Li, 2021; Di Fazio & Gullerova, 2023; Drino et al., 2023). tRNA-derived stress-induced RNAs (tiRNAs), comprising 5′ and 3′ tiRNAs (also known as tRNA halves or tRHs, as shown in Fig. 1), are 29–50 nucleotides long. They are produced by the cleavage of the anticodon loop in mature tRNAs by ribonuclease A (RNase A) or ribonuclease T2 (RNase T2) (Megel et al., 2019; Tan, Tan & Duan, 2019; Magee & Rigoutsos, 2020; Su et al., 2020; Ma, Liu & Cao, 2021). The utilization of RNase A vs. RNase T2 for generating tsRNAs from mature tRNAs likely depends on subcellular localization, species-specific expression patterns, and stress-induced signaling pathways. RNase A is a cytosolic enzyme that typically cleaves tRNAs in the cytoplasm under general stress conditions (e.g., amino acid deficiency, ultraviolet irradiation, or heat shock) to produce 5′- and 3′-tiRNAs. In contrast, RNase T2 is associated with acidic vacuoles in plants and may be preferentially activated under osmotic or oxidative stress that disrupts vacuolar homeostasis. For example, Arabidopsis RNS2 (a RNase T2 homolog) localizes to the endoplasmic reticulum and vacuoles and is induced by ABA, suggesting its role in stress-specific tsRNA biogenesis (Hillwig et al., 2011). Further experimental validation (e.g., analyzing enzyme expression patterns under different stresses) would clarify these regulatory mechanisms. The biogenesis of tiRNAs originates from a non-Dicer-dependent pathway, where Dicer belongs to the RNase III family of endoribonucleases that specifically recognize and cleave double-stranded RNA. The cleavage events occur under various stress conditions, including amino acid deficiency, phosphate starvation, ultraviolet irradiation, heat shock, hypoxia, oxidative damage, and viral infection. Under typical growth conditions, tiRNAs are rarely generated and predominantly localized in the cytoplasm, with minimal presence in the nucleus and mitochondria (Li, Xu & Sheng, 2018).

Table 1 tsRNAs species.

Category	Subcategory	Biogenesis	Characteristics	
tiRNAs	5′ tiRNA	Ribonuclease A (RNase A) or Ribonuclease T2 (RNase T2) cleave the anticodon loop of mature tRNA	The sequence between the 5′ end of mature tRNA and the cleavage site of the anticodon loop	
3′ tiRNA	The sequence between the 5′ end of mature tRNA and the cleavage site of the anticodon loop	
tRFs	5-tRFs	Dicer or RNase T2 cleaves the D-loop of mature tRNA	The sequences at the 5′ end of mature tRNA can be further classified into 5a-tRFs (approximately 15 nt), 5b-tRFs (approximately 22 nt), and 5c-tRFs (approximately 30 nt), based on their lengths.	
3-tRFs	Dicer or RNase A cleaves the T-loop of mature tRNA	The sequences at the 3′ end of mature tRNA, often including the terminal CCA sequence, can be further divided into 3a-tRFs (approximately 18 nt) and 3b-tRFs (approximately 22 nt), based on their lengths.	
1-tRFs	RNase Z cleaves the 3′ tail of pre-tRNA	These are single-stranded, ranging from 20 to 40 nt in length, with a 3′ poly-U tail, and are distributed in both the nucleus and cytoplasm.	
2-tRFs	Originating from tRNAGlu, tRNAAsp, tRNAGly and pre-tRNATyr	There are only four types of 2-tRFs, which contain a complete anticodon arm and anticodon loop.	
i-tRFs	Originating from the interior of mature tRNA	Starting from the 5′ end of mature tRNA, they encompass the anticodon loop and a portion of the D/T loop.	

Figure 1 Classifies the types of tsRNAs based on their size and sequence location within the tRNA structure (Zhu, Ow & Dong, 2018; Ma, Liu & Cao, 2021; Tan, Tan & Duan, 2019; Megel et al., 2019; Li, Xu & Sheng, 2018; Li et al., 2021).

RNase Z cleaves the 3′ end of pre-tRNA to produce 1-tRF. 2-tRF, which retains a complete anticodon loop and stem, is generated by an unidentified ribonuclease. 3-tRF refers to the 3′ terminal sequence produced by the cleavage of the T loop in mature tRNA by Dicer or RNase A. 5-tRF is the 5′ terminal sequence resulting from Dicer or RNase T2 cleavage of the D loop in mature tRNA. 5′-tiRNA and 3′-tiRNA are derived from the respective ends of mature tRNA, extending to the anticodon loop cleavage site by RNase A or RNase T2. i-tRF is generated by an unidentified ribonuclease from the internal region of mature tRNA, including the anticodon loop and parts of the D/T-loop.

tRFs are a subclass of tsRNAs, ranging from 16 to 28 nucleotides in length, originating from primary or mature tRNAs and encompassing various subtypes. tRF biogenesis includes Dicer-dependent and non-Dicer-dependent pathways (Fig. 1) (Li, Xu & Sheng, 2018; Li et al., 2021). Table 1 presents the identified five tRF subtypes, 5-tRFs, 3-tRFs, 1-tRFs, 2-tRFs, and inter-tRFs (i tRFs), along with their characteristics (Pan, Han & Li, 2021). 5-tRFs are generated by the cleavage of the tRNA D-loop by Dicer or RNase T2, often with an adenosine as their 3′ end; 5-tRFs can be further classified on the basis of length as 5a tRFs (approximately 15 nt), 5b tRFs (approximately 22 nt), and 5c tRFs (approximately 30 nt) (Kumar, Kuscu & Dutta, 2016; Tan, Tan & Duan, 2019; MacIntosh & Castandet, 2020; Ma, Liu & Cao, 2021). 3-tRFs, typically 18 or 22 nucleotides long and often containing a CCA tail, are generated by Dicer or RNase A cleavage of the T-loop (Maraia & Lamichhane, 2011; Li et al., 2012; Kumar et al., 2015). 1-tRFs are produced by RNase Z cleaving the 3′ trailer of pre-tRNAs, typically just beyond the 3′ end of mature tRNAs, excluding the 5′-CCA-3′ sequence. 1-tRFs have a poly-U at their 3′ ends (Babiarz et al., 2008; Su et al., 2020). 2-tRFs originate from the tRNAs recognizing the codon for specific amino acids—tRNAGlu, tRNAAsp, tRNAGly, and pre-tRNATyr-and encompass the entire anticodon loop (Goodarzi et al., 2015). i tRFs originate from the internal regions of mature tRNAs, encompassing the anticodon loop and portions of the D-loop and T-loop (Su et al., 2020; Li et al., 2021). The specific ribonucleases involved in the formation of 2-tRFs and i tRFs remain unidentified (Li et al., 2021). To address the inconsistent naming of tsRNAs, a standardized naming convention X-tsRNAAA-NNN is proposed, where tsRNA indicates the type (tiRNA or tRF), X denotes the subtype (1, 2, 3, or 5 based on tRNA location), AA is the three-letter amino acid code, and NNN is the tRNA anticodon (Kumar et al., 2015; Guan & Grigoriev, 2020; Li et al., 2021; Zuo et al., 2021; Holmes et al., 2023). For instance, 5′ tiRNA and 3a tRF originating from tRNAGlu-CUC should be designated as 5′-tiRNAGlu-CUC and 3a-tRFGlu-CUC, respectively.

Subcellular localization of tsrna biogenesis

Due to differences in experimental methods and species used, the subcellular localization of tsRNAs remains controversial. Ma, Liu & Cao (2021) pointed out that determining the location of tsRNA biogenesis within the cell would provide valuable insights into the functions of tsRNA. Cytoplasmic tRNAs can be transported to vacuoles for cleavage by RNase T2. Megel et al. (2019) found that the RNase T2 family plays an important role in the biogenesis of tsRNAs in yeast (Saccharomyces cerevisiae), plants, and human cells. RIBONUCLEASE 1 (RNS1), RNS2, and RNS3 are members of the RNase T2 family in Arabidopsis (Arabidopsis thaliana) and are acidic ribonucleases mainly localizing to acidic vacuoles. Alternatively, cytoplasmic RNase T2 can generate many tsRNAs in the cytoplasm (Ma, Liu & Cao, 2021; Ma et al., 2023; Liang et al., 2024). Arabidopsis RNS2 has been identified in the endoplasmic reticulum (ER) (Hillwig et al., 2011), suggesting that tsRNA biogenesis may occur in the ER and be linked to its associated polyribosomes, akin to the established functions of miRNAs in the ER, which involve regulating protein folding and quality control, modulating lipid metabolism, and controlling ER stress responses (Yang et al., 2021).

Biological functions of tsRNAs

Various studies indicate that tsRNAs interact with proteins to modulate gene expression at transcriptional, post-transcriptional, and translational stages via multiple mechanisms (Figs. 2 and 3) (Li, Xu & Sheng, 2018; Li et al., 2021; Park & Kim, 2018; Shi, Zhou & Chen, 2022; Fu et al., 2023). tRFs and tiRNAs perform diverse biological functions. For example, tsRNAs can function as miRNAs under specific conditions, such as when bound to AGO proteins, to repress target mRNA translation or induce degradation (Gebetsberger et al., 2012; Sobala & Hutvagner, 2013). Additionally, they may displace the translation initiation factor eIF4G from mRNAs to directly inhibit protein synthesis (Gebetsberger et al., 2012). Furthermore, tsRNAs interact with proteins like YBX1/YB-1 to modulate mRNA stability and localization (Sobala & Hutvagner, 2013), while certain tsRNAs can bind cytochrome C to regulate apoptosis pathways (Saikia et al., 2014). Under oxidative or ER stress, they promote stress granule (SG) assembly and sensitize cells to p53-dependent apoptosis (Hanada et al., 2013). Lastly, specific tsRNAs (e.g., Arabidopsis tsRNA166) act as paternal epigenetic regulators by guiding DNA methylation in sperm, thereby modifying offspring transcriptional programs (Chen et al., 2016). Furthermore, RFs exhibit functions such as Dicer-dependent biogenesis, RISC formation with Argonaute proteins, and RNA silencing (Venkatesh, Suresh & Tsutsumi, 2016). tRFs or their analogs displace the mRNA-binding protein YBX1 from the 3′ untranslated region (3′ UTR) of mRNAs from many oncogenic genes, thus rendering them unstable and inhibiting the invasiveness of cancer cells (Goodarzi et al., 2015).

Figure 2 The versatile roles and operational mechanisms of tsRNAs (Li, Xu & Sheng, 2018).

tsRNAs regulate various biological processes, including gene expression, translation initiation and elongation, stress granule formation, ribosome biogenesis, and transgenerational inheritance. LTR, long terminal repeat; RISC, RNA-induced silencing complex; PBS, primer binding sites, YB1, Y-box binding protein 1.

Figure 3 A proposed model for the role of tsRNAs in modulating gene expression during stress responses (Park & Kim, 2018).

Mature tRNAs that have undergone methylation by Dnmt2 and Nsun2 are involved in translation processes.Conversely, unmethylated mature tRNAs or pre-tRNAs are processed into various categories of tsRNAs (Thompson & Parker, 2009; Schaefer et al., 2010; Blanco et al., 2014; Kumar, Kuscu & Dutta, 2016). Abiotic stress induces tsRNA production, which regulates gene expression at transcriptional, post-transcriptional, and translational levels.

tsRNAs suppress transcription

PIWI proteins interact with piRNAs and belong to the AGO protein family. PIWIs inhibit the transcription of genes present on transposons (Le Thomas et al., 2013). Notably, Keam et al. (2014) demonstrated that tRFs can interact with PIWI proteins in a piRNA-like manner, repressing transcription in somatic cells (Fig. 2). This interaction was specifically observed in human cells using tRFs derived from mature tRNAs (Keam et al., 2014). While tRF-PIWI associations have been implicated in transcriptional suppression in mammals, similar observations in plants remain to be demonstrated.

tRFs regulate mRNA stability

As tRFs bind to AGO proteins, one of the components of the RISC, tRFs have been predicted to downregulate the expression of target genes. As sncRNAs shorter than 30 nt, tRFs have similar functions to miRNAs via direct binding. Similar to the AGO–miRNA interaction, the target sequences of tRFs may thus be predicted based on the tRF sequences (Sharma et al., 2016; Schorn et al., 2017). 3-tRFs from tRNAGly-GCC in mature B lymphocytes and tRNALeu-CAG in lung cancer cells exhibit miRNA-like structures and functions, enabling them to cleave mRNAs with partial complementarity and/or inhibit protein translation (Maute et al., 2013; Shao et al., 2017). Kumar et al. (2014) found that tRFs in different cell types preferentially bind to different AGO proteins, with some preferring to bind to AGO1, AGO3, and AGO4 but not AGO2. 5a tRFs selectively bind to AGO1 or AGO2 (Martinez, Choudury & Slotkin, 2017), while 3b tRFs bind to AGO2 (Maute et al., 2013). In Arabidopsis, 5-tRFs were also shown to bind to AGOs other than AGO2, while 3b tRFs have a strong binding preference for AGO2 (Loss-Morais, Waterhouse & Margis, 2013). Haussecker et al. (2010) reported that 3-tRFs and 1-tRFs regulate gene expression by competitively binding to AGO family members, influencing the degree to which their target transcript abundance is reduced. The 1-tRFSer-UGA, generated from pre-tRNASer-UGA cleavage, associates with AGO3 and AGO4 without silencing gene expression like miRNAs. Consequently, tRFs can modulate gene expression by interacting with AGO proteins via both traditional and alternative miRNA/siRNA pathways (Fig. 2).

Importantly, the specificity of the AGO–tRF interaction depends on the amino acid transported by the corresponding tRNA (Park & Kim, 2018). Li et al. (2012) showed that AGO2 binds to 3b-tRFLeu-CAG more strongly than to 3b-tRFHis-GUG. Similar results were reported in rice (Oryza sativa), where AGO1 interacts more strongly with 5a-tRFArg-CCU than with 5a-tRFAla-AGC (Alves et al., 2017). Cognat et al. (2017) found that some stress conditions enhance the interaction between tRFs and AGOs in Arabidopsis, with UV light irradiation enhancing the binding of 5a-tRFGly-UCC to AGO1. By contrast, tiRNAs (30–40 nt) may not bind to AGO proteins and function through the miRNA/siRNA pathway, as AGOs bind to small RNAs of 20–24 nt in length (Dueck et al., 2012). 2-tRFs can bind competitively to YBX1, disrupting its interaction with oncogenic mRNAs, thereby reducing their stability and inhibiting human breast cancer cell metastasis (Fig. 2) (Goodarzi et al., 2015).

There is clearly a need for a more in-depth exploration of the mechanism by which tsRNAs regulate gene expression at the post-transcriptional level, the identification of the endogenous target mRNAs of tRFs, and the construction of their corresponding regulatory expression networks.

tsRNAs regulate translation

tsRNAs can positively or negatively regulate protein translation (Ma, Liu & Cao, 2021). tiRNAs (primarily 5 tiRNAs) can decrease the overall translation rate by 10–15% (Yamasaki et al., 2009). Specific 5′ tiRNAs from tRNAAla and tRNACys can form G-quadruplex-like structures (G4 motifs). These structures competitively bind to eIF4G or eIF4A in the translation initiation complex, inhibiting cap-dependent mRNA translation without impacting IRES-mediated protein translation, which supports cell survival and prevents apoptosis. An internal ribosome entry site (IRES) is a cis-acting element within the mRNA molecule (not at the 5′ end) that can be recognized by ribosomes to initiate translation (Ivanov et al., 2011; Ivanov et al., 2014). Most proteins involved in cell survival and with anti-apoptosis function are translated through the IRES pathway. Under stress conditions, tiRNAs may diminish energy consumption by the cell via selectively inhibiting the translation of housekeeping genes while not affecting the formation of pro-survival proteins, thus exerting a protective effect. The results indicate that tiRNA formation aims to regulate translation output during stress rather than altering the levels of functional mature tRNAs. Five tiRNAs can interact with the translation repressor YBX1 to enhance stress granule assembly via an eIF2α phosphorylation-independent pathway, thus boosting cellular stress resistance and survival (Fig. 2) (Lyons et al., 2016). Therefore, tiRNAs are important regulatory factors for cells under stress conditions. Zhang, Sun & Kragler (2009) found that tsRNAs in pumpkin (Cucurbita pepo) phloem sap may inhibit translation by interfering with ribosomal activity. Recent studies in Arabidopsis also showed that, similar to human cells, tsRNAs in plants can interact with polysomes to inhibit translation, and the three nucleotides G18, G19, and A16 of tRNAs are crucial for tsRNA-mediated translation inhibition (Lalande et al., 2020). Gebetsberger et al. (2017) determined that under stress conditions, 5-tRFVal produced by the halophilic archaeon Haloferax volcanii could bind to the small ribosomal subunit, inhibiting the formation of translation initiation complexes between mRNAs and ribosomes, thus leading to overall translation attenuation. Furthermore, archaeal 5-tRFVal can also inhibit protein production in eukaryotes and bacteria, suggesting a conserved function for 5-tRFVal across species.

Conversely, several studies have shown that tsRNAs can also enhance translation. For example, Kim et al. (2017) demonstrated that a 3 tRF derived from tRNALeu-CAG in human cells can bind to two mRNAs encoding ribosomal proteins and promoted their translation.

tRFs regulate ribosome biogenesis

Recent research suggests that tRFs play a crucial role in regulating ribosome biogenesis, encompassing both ribosomal RNA (rRNA) and ribosomal protein formation (Li, Xu & Sheng, 2018). In research on the unicellular ciliate Tetrahymena thermophila, tRFs were identified in the pre-rRNA splicing complex (Couvillion et al., 2012). 3-tRFs specifically bind to Twi12, a member of the AGO/PIWI protein family, recruiting Tan1 and exonuclease Xrn2 to form the TXT complex. This complex cleaves and processes pre-rRNAs to facilitate rRNA synthesis (Fig. 2) (Couvillion et al., 2012). However, whether tRFs can enhance rRNA processing in multicellular organisms remains to be investigated. In mammalian cells, 3-tRFs originating from tRNALeu-CAG can interact with mRNAs of ribosomal proteins ribosomal protein S28 and ribosomal protein S15, enhancing their translation (Kim et al., 2017).

As a new epigenetic regulatory factor

Transposons, mobile DNA sequences that can damage the host genome, are often transcriptionally repressed by epigenetic marks like DNA methylation and histone modifications (Slotkin & Martienssen, 2007). Schorn et al. (2017) discovered that 3-tRFs of varying lengths (18 and 22 nucleotides) inhibit the propagation of long terminal repeat (LTR) transposons via two distinct mechanisms: the 18-nt 3-tRF blocks reverse transcription, while the 22-nt 3-tRF functions similarly to miRNA in post-transcriptional silencing (Fig. 2). In Arabidopsis, 5-tRFs can regulate genome stability by targeting transcripts derived from transposons in a manner similar to miRNAs (Martinez, Choudury & Slotkin, 2017). tiRNAs are the predominant small RNA class in mature mouse sperm (Peng et al., 2012). During fertilization, tiRNAs accumulate and can modify the early embryo transcriptome, affecting the transcription of genes associated with metabolic pathways (Sharma et al., 2016; Chen et al., 2016). These changes occur regardless of DNA methylation status, indicating that tiRNAs might function as a novel epigenetic regulator affecting offspring phenotype. These findings open a new chapter in RNA-mediated epigenetic regulation.

Regulation of plant gene expression by tsRNAs in response to abiotic stress

Although the biogenesis and precise functions of tsRNAs in plants remain unclear, studies have demonstrated that specific tsRNAs accumulate in response to distinct stress conditions (Table 2). For example, Arabidopsis DCL4-dependent tsRNAs are induced under heat stress (Thompson et al., 2008), while tRFs derived from mature tRNAs accumulate in human cells during oxidative stress (Chen et al., 2024). Notably, the biogenesis of some tsRNAs is regulated by stress-responsive pathways, such as the RNA-directed DNA methylation (RdDM) pathway, which is involved in plant defense and stress adaptation (Wang et al., 2016). However, direct evidence linking tsRNA accumulation to specific plant stress conditions remains limited, with most studies focusing on animal models or non-stress-related contexts (e.g., transgene silencing).

Table 2 Various tsRNAs generated in plants in response to abiotic stresses.

Plant	Abiotic stresses	TsRNAs	
Arabidopsis thaliana	Oxidative stress	tiRNAHis-GUG, tiRNAArg-CCU, tiRNATrp-CCA; 5-tRFArg-UCG, 3-tRFTyr-GUA	
Phosphorus starvation	5-tRFAsp-GUC, 5-tRFGly-UCC	
Drought stress	5-tRFArg-UCG, 5-tRFArg-CCU, 5-tRFGly-UCC, 5-tRFAla-AGC	
wounding stress	5-tRFAla-CGC/TGC, 5-tRFAla-AGC, 5-tRFAla-TGC/AGC, 5-tRFAla-CGC	
Ultraviolet radiation	5-tRFGly-GCC, 5-tRFGly-UCC, 5-tRFPro-UGG, 5-tRFVal-AAC	
Arabidopsis thaliana, Triticum aestivum	Salt stress	5-tRFAla-AGC; 3-tRFVal-CAC, 3-tRFThr-UGU, 3-tRFTyr-GUA, 3-tRFSer-UGA	
Triticum aestivum	Heat stress	3-tRFThr-UGU, 3-tRFTyr-GUA, 3-tRFSer-UGA	
Oryza sativa	Cold stress	5-tRFArg-CCU	

For example, UV-B light treatment promoted the formation of tRFGly-UCC, which can be detected among components immunoprecipitated with AGO1 (Cognat et al., 2017). Similarly, drought stress promoted the formation of 5a tRFArg and enhanced its binding to AGO1 (Alves et al., 2017). So far, genes involved in the biogenesis of plant tsRNAs under abiotic stress include DICER-LIKE 1 (DCL1) and RNS1 (Park & Kim, 2018). The significant reduction in the abundance 5a tRFAla in dcl1-11 mutant pollen confirms DCL1-dependent tRF biogenesis (Martinez, Choudury & Slotkin, 2017). The expression of RNS1, whose encoded protein participates in the non-Dicer-dependent pathway of tsRNA biogenesis, is induced by treatment with the phytohormone ABA, and the levels of tsRNAs increase in RNS1-overexpressing plants relative to wild-type plants (Luan et al., 2020; Hillwig et al., 2011; Alves et al., 2017), ABA may induce tsRNA production through the RNS1-dependent pathway. In addition, the trdmt1 and trm4 mutants of Arabidopsis lack tRNA methylation, suggesting that the methyltransferases TRDMT1 (homolog of human Dnmt2) and tRNA METHYLTRANSFERASE 4 (TRM4, homolog of human Nsun2) in plants are involved in the formation of tsRNAs (Burgess, David & Searle, 2015), various enzymes modifying tRNAs in Arabidopsis may also participate in the biosynthesis of tsRNAs (Chen, Jäger & Zheng, 2010). Under abiotic stress conditions (e.g., heat stress or oxidative stress), tRNA methylation levels dynamically change, which may influence the biogenesis of tsRNAs through altered tRNA stability or processing efficiency. Wang et al. (2017) pointed out that the modification levels of tRNAs in plants exposed to salinity stress did indeed change. AGO proteins were shown to bind to tRFs upon non-biotic stress conditions; in addition to methylation catalyzed by TRDMT1 and TRM4, other new tRNA modifications may be involved in the formation of tsRNAs in plants under different stresses (Park & Kim, 2018).

In the context of transposon suppression, the AGO1-tRF complex plays a critical role in maintaining genome integrity under stress conditions. Transposon activation is often associated with stress responses, such as DNA damage or pathogen infection, which can lead to genomic instability if not tightly controlled. For instance, 5a tRF interacts with AGO1 in Arabidopsis pollen and inhibits the expression of the transposon Athila6A at the post-transcriptional level in a DCL1-dependent manner (Martinez, Choudury & Slotkin, 2017). tsRNAs are also important in regulating the activity of plant transposons, thus protecting genome integrity (Ma, Liu & Cao, 2021). Stress-induced metabolic changes, such as lipid peroxidation and iron accumulation, can modulate the activity of transposon suppression pathways. For example, the activation of ferroptosis, a stress-induced form of cell death, has been shown to involve the regulation of small RNA biogenesis factors, highlighting the interconnectedness of stress responses and genome defense mechanisms (Shen et al., 2020). Since TRDMT1 plays an important role in tRF formation, the interaction between TRDMT1 and HISTONE DEACETYLASE 2C (HD2C) in the nucleus suggests that tRFs can inhibit gene expression by affecting histone modifications, at least in Arabidopsis (Song et al., 2010). HD2C-overexpressing plants exhibited a decreased sensitivity to ABA and lower expression of ABA-responsive genes, while hd2c mutant plants showed the opposite phenotypes (Sridha & Wu, 2006; Luo et al., 2012). The close association between an increase in specific tsRNAs levels and abiotic stress signaling transduction indicates that tsRNAs can repress the expression of stress-related genes at transcriptional, post-transcriptional, and translational levels (Fig. 3) (Park & Kim, 2018). From tRNAs-derived fragments (tsRNAs) that can enhance translation through sequence-specific interactions with RNA-binding proteins (RBPs) to tsRNAs that inhibit normal growth by repressing key developmental genes, tsRNAs can therefore finely regulate gene expression in plants responding to stress in a sequence-specific manner (Shen et al., 2020).

In summary, tsRNAs may be involved in many plant responses to stress (Ma, Liu & Cao, 2021; Li et al., 2023). However, the molecular mechanisms by which tsRNAs participate in abiotic stress are still unclear. Does stress alter the pools of plant tRNAs, leading to the formation of different types of tsRNAs? Do some plant tsRNAs regulate the expression of stress resistance genes, as miRNAs would? Investigating these mechanistic questions will enhance our understanding of tsRNAs’ role in plant stress responses.

Research methods of tsRNAs in plants

With the discovery and identification of tsRNAs in an increasing number of species, in-depth investigation of their biological functions and corresponding regulatory mechanisms has become a new research hotspot. Currently, the main research methods for tsRNAs involve screening and identifying different types of tsRNAs through high-throughput sequencing, followed by determination of their target molecules (targets), conducting in vivo and/or in vitro functional validation of these targets, and constructing their regulatory networks. Creating a detailed description and annotation of the global tsRNA library is a daunting task, primarily due to technical hurdles posed by the presence of modified nucleotides in tsRNAs and the inadequate ability to purify these modified forms. These modifications in small RNAs can disrupt the reverse transcription process, resulting in either the cessation of transcription or the introduction of sequence errors during high-throughput RNA sequencing (Drino et al., 2020; Zuo et al., 2021; Chery & Drouard, 2023; Scacchetti et al., 2023; Liang et al., 2024).

As with tRNAs, the multiple modifications present on tRFs and tiRNAs make them unsuitable for direct standard small RNA-seq. Indeed, abundant base modifications can severely hinder the reverse transcription step during library generation, and the various specific terminal modifications of tRFs and tiRNAs prevent them from being connected to an adapter. To obtain a more accurate view of tRF and tiRNA expression profiles, purified RNA requires processing prior to library construction, including deamination (removal of 3′ amino acids), dephosphorylation (conversion of 3′ cyclic phosphate to hydroxyl groups), terminal phosphorylation (phosphorylation of 5′ hydroxyl groups), and demethylation (to prevent interruption of reverse transcription). Traditional small RNA sequencing (sRNA-seq) can only capture tsRNAs with 3′ hydroxyl (3′ OH) termini (Alves et al., 2017; Cognat et al., 2017), while RNA sequencing with 3′ cyclic phosphate termini (cP-RNA-seq) only captures tsRNAs with 3′ cP termini (Honda, Morichika & Kirino, 2016). Recently developed techniques such as tRNA-seq (Ma et al., 2021a; Zheng et al., 2015), panoramic RNA display by overcoming RNA modification aborted sequencing (PANDORA-seq) (Shi et al., 2021), AlkB-facilitated RNA methylation sequencing (ARM-Seq) and CPA-seq can analyze sRNAs with different termini and nucleoside methylations, facilitating the identification of numerous tsRNAs in mammals and plants (Cozen et al., 2015; Wang et al., 2021). Megel et al. (2019) recently demonstrated that RNase T2 is the ribonuclease responsible for the biogenesis of tiRNAs in Arabidopsis, hydrolyzing tRNAs to form tsRNAs (3′ cP tsRNAs) with 3′ cyclic phosphate termini. The cP-RNA-seq methodology is therefore required to access and identify these tsRNAs in plants (Honda, Morichika & Kirino, 2016). The tRNA-seq and Y-shaped adapter-ligated mature tRNA sequencing (YAMAT-seq) methods were recently employed to systematically identify tRNAs and tsRNAs in Arabidopsis, rice, maize (Zea mays), soybean (Glycine max), wheat (Triticum aestivum), barley (Hordeum vulgare), and sorghum (Sorghum bicolor), providing an in-depth analysis of the composition of plant tsRNA populations (Ma et al., 2021b; Shigematsu et al., 2017). The 5′ tiRNAs (i.e., 5′ tRHs) were the most abundant small RNAs, and the accumulation of tsRNAs was highly tissue-specific. Additionally, a class of tiny tsRNAs less than 18 nt in length was widely present in different plant species, with 13-nt and 16-nt tiny tsRNAs being the major subtypes in Arabidopsis. Gu et al. (2022) recently developed a new method for small RNA library construction and sequencing based on the RtcB ligase from Escherichia coli specifically for tsRNAs (RtcB sRNA-seq). This method can simultaneously capture and quantitatively distinguish small RNAs with 3′ hydroxyl and 3′ phosphate/cyclic phosphate termini, allowing the analysis of the proportion of tsRNAs with 3′ hydroxyl and 3′ phosphate/cyclic phosphate termini. RtcB sRNA-seq allowed the systematic identification and analysis of 5′ tiRNAs and 5-tRFs in Arabidopsis seedlings, revealing that 80 tRNAs can produce nearly a thousand 5′ tiRNAs and 5-tRFs, most of which have 3′ cyclic phosphate termini. Among them, 5′ tiRNAs and 5-tRFs derived from tRNAAla, tRNAGly, and tRNAGlu had the highest abundance. 5-tRFAla was shown to negatively regulate plant antifungal defense responses by downregulating the expression of its target gene Cytochrome P450 71A13 (CYP71A13) and the biosynthesis of phytoalexins. However, the RtcB sRNA-seq method has the disadvantage that it cannot capture tsRNAs with 5′ hydroxyl termini, including 3′ tiRNAs and 3-tRFs (Gu et al., 2022).

In addition to optimized sequencing methods, several bioinformatics tools and databases have been reported for analyzing plant tsRNAs, including the Arabidopsis-specific tsRNA database tRex (Thompson et al., 2018), the plant tsRNA database PtRFdb (collecting information on over 5,000 tsRNAs from 10 plant species, including Arabidopsis, purple false brome (Brachypodium distachyon), soybean, alfalfa (Medicago sativa), Physcomitrium patens, black cottonwood (Populus trichocarpa), sorghum, grape (Vitis vinifera), and maize) (Gupta et al., 2018), the tsRBase database (collecting information on over 120,000 tsRNAs from 20 plant species, including their expression patterns, functions, target molecules, and other details) (Zuo et al., 2021), and tRFanalyzer (creating tsRNA databases for Arabidopsis and rice, allowing queries and analysis of the spatiotemporal accumulation of tRNAs and tsRNAs) (Ma et al., 2021b). Classical molecular biology experimental techniques are also being used to dissect the interactions between tsRNAs and nucleic acids and proteins. In summary, advancements in sequencing technologies and online databases have enabled the quantification of tsRNA populations in plants, aiding in the understanding of their functions.

Conclusions and outlook

tsRNAs, comprising two major classes of tRNA-derived small molecules (tiRNAs and tRFs), have attracted considerable attention due to their unique and diverse biological functions. tRNA precursors or mature tRNAs can form tsRNAs through both Dicer-dependent and Dicer-independent pathways (Figs. 1 and 3) (Li et al., 2012; Park & Kim, 2018). The generation of specific tsRNAs under different stress conditions may be determined by the amino acids transported by each tRNA or different tissue types (Hsieh et al., 2009; Yamasaki et al., 2009; Alves et al., 2017; Martinez, Choudury & Slotkin, 2017). tRNA modifications, such as methylation regulated by Dnmt2 and Nsun2 in human cells or TRDMT1 and TRM4 in plants, play a crucial role in the biogenesis of tsRNAs (Burgess, David & Searle, 2015; Park & Kim, 2018). Cellular tsRNA levels change under stress conditions like oxidative stress, nutrient deficiency, or heat stress, suggesting their specific role in stress tolerance responses (Thompson & Parker, 2009; Emara et al., 2010; Schaefer et al., 2010; Li, Xu & Sheng, 2018). tsRNAs modulate stress signaling by inhibiting gene expression at various stages, including transcription, post-transcription, and translation. Under stress, plants utilize tsRNAs to swiftly reduce gene expression, thereby minimizing non-essential cellular activities and enhancing survival (Park & Kim, 2018). Identifying the target genes or transcripts of specific tsRNAs will help us understand the mechanisms of tsRNA-mediated downregulation of target gene expression, elucidating the role of tsRNAs in the regulation of stress responses.

The complement and abundance of tRFs and tiRNAs are highly dependent on cell type (Magee & Rigoutsos, 2020). The content of tRNAs, tRFs, and tiRNAs is sometimes even higher than that of miRNAs (Megel et al., 2019; Ma, Liu & Cao, 2021; Pan, Han & Li, 2021). Although the screening of biomarkers currently mainly focuses on miRNAs, the high abundance and stability of tRFs and tiRNAs, their widespread involvement in different areas of biology, and their strong discriminatory ability under different stress conditions have opened up broad prospects for the development of tRFs and tiRNAs as biomarkers.

While extensive research has delved into the biogenesis and functions of tsRNAs in mammalian cells, the study of plant tsRNAs is still in its infancy, with many questions remaining to be addressed. First, standardized protocols for the construction and bioinformatics analysis of tsRNA sequencing libraries need to be established and/or improved. Most current studies only apply library preparation schemes designed for miRNA-based sequencing to tsRNAs, for example by selecting small RNAs of 18–28 nt in length, which excludes some tsRNA subclasses, such as tiRNAs of 30–40 nt in length (Li et al., 2021). Adjusting the size selection of small RNAs during library preparation will substantially affect tsRNA identification. Second, the biogenesis of plant tsRNAs is still unclear. Since tsRNAs can form independently of DCL or RNS, it is reasonable to speculate that other ribonucleases are also involved in tsRNA biogenesis (Zhu, Ow & Dong, 2018). Under a specific set of conditions or at a particular developmental stage, only a few tsRNA species are enriched, suggesting that tsRNA biogenesis is highly specific and tightly regulated. The molecular regulatory mechanisms of tsRNA biogenesis remain unknown. Arabidopsis RNS2 is mainly localized in the vacuole, but it is unclear whether the tsRNA pool in plants resides in the vacuole. Are tsRNAs stable in cells? Where are they degraded? These questions require further investigation. Third, given the significant influence of stress on crop yield, deciphering the functions of tsRNAs related to stress responses is particularly valuable for agriculture. However, our current understanding of the physiological functions of specific tsRNAs is very limited in plants. Various stresses can induce changes in plant tsRNAs, but it is unclear whether these changes can enhance plant stress tolerance and whether specific tsRNAs can confer stress tolerance to plants similar to miRNAs (Ma, Liu & Cao, 2021). tsRNA-mediated regulation of gene expression may be part of the plant defense system in response to abiotic and biotic stresses (Zhu, Ow & Dong, 2018). Experiments aimed at assessing the stress response of tsRNA biogenesis mutants could help answer these questions, but genetically dissecting the function of a specific tsRNA will be more technically challenging. New methods for overexpressing or knocking out individual tsRNAs without interfering with the tRNAs and other small RNAs they are derived from still need to be developed.

We specially thank the editing assistance of Plant Editors for polishing the language of our manuscript.

Additional Information and Declarations

Competing Interests

The authors declare that they have no competing interests.

Author Contributions

Chunmei Li conceived and designed the experiments, performed the experiments, prepared figures and/or tables, and approved the final draft.

Jing Zhu performed the experiments, prepared figures and/or tables, and approved the final draft.

Han Jin performed the experiments, prepared figures and/or tables, and approved the final draft.

Haotian Feng performed the experiments, prepared figures and/or tables, and approved the final draft.

Haimin Zhuang performed the experiments, prepared figures and/or tables, and approved the final draft.

Zijun Du performed the experiments, prepared figures and/or tables, and approved the final draft.

Guolin Zhu performed the experiments, prepared figures and/or tables, and approved the final draft.

Haiyang He performed the experiments, prepared figures and/or tables, and approved the final draft.

Fuyang Ye performed the experiments, prepared figures and/or tables, and approved the final draft.

Zhaohui Mo performed the experiments, prepared figures and/or tables, and approved the final draft.

Qingtao Hu performed the experiments, prepared figures and/or tables, and approved the final draft.

Zhenbang Chen performed the experiments, prepared figures and/or tables, and approved the final draft.

Kai Liu conceived and designed the experiments, analyzed the data, authored or reviewed drafts of the article, and approved the final draft.

Xiaorong Wan conceived and designed the experiments, authored or reviewed drafts of the article, and approved the final draft.

Data Availability

The following information was supplied regarding data availability:

This is a literature review.

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
