# Peer review of "Regulation of plant gene expression by tsRNAs in response to abiotic stress"

_PeerJ, doi:10.7717/peerj.19487_

## Round 0.1 · original submission · Major Revisions

Dear Dr. Wan,

Based on the reviewers' comments, the manuscript requires major revisions before it can be considered for acceptance. The key reasons for this decision are:

Strengths of the Manuscript:
The manuscript is well-written and addresses a relevant and current topic in plant research.
The literature is sufficient, although more recent references from 2024 and 2025 are recommended.
The research methodology is clearly described and appropriate.
The manuscript provides valuable insights into the role of tsRNAs in plant stress responses, a relatively underexplored field.
Major Concerns and Revisions Required:
Structural and Organizational Issues:

The introduction lacks depth and needs additional references to better define key concepts such as post-transcriptional modifications, signaling pathways, and miRNA functions in stress responses.
The review structure needs reorganization, particularly aligning the section order with the conclusion for better readability.
Several sections, particularly in subcellular localization and biological functions of tsRNAs, are fragmented and lack logical flow.
Clarification and Elaboration of Key Concepts:

More specific examples and citations are needed for:
Post-transcriptional modifications of tsRNAs (Line 37-38).
Stress response pathways in plants (Line 39-41).
Functions of miRNAs in the endoplasmic reticulum (Line 124-126).
The distinction between tiRNAs and tRFs should be consistently applied throughout the text.
Additional explanations of plant-specific tsRNA functions, rather than relying heavily on mammalian models.
Improving Methodological Depth:

Reviewers suggest including in silico analysis from the tsRNA database to enhance the manuscript's contribution beyond summarizing existing literature.
References to inconsistent naming conventions should be properly cited (e.g., Kumar et al., 2014; Pliatsika et al., 2017).
Figures and Presentation:

Figures should have a white background instead of sky blue for better readability.
The addition of a small RNA machinery overview figure is suggested to help readers unfamiliar with the topic.
Conclusion and Future Perspectives:

The conclusion is too long and should be broken down, with some content moved to relevant sections.
The future perspectives lack depth—authors should propose specific methods or experiments to address the gaps identified in their review.
Decision: Major Revisions
The manuscript presents valuable content but requires significant restructuring, expansion of the introduction, clearer organization, and additional analyses. Authors should carefully address the detailed reviewer comments, improve the clarity and logical flow, and include more recent references and potential analyses. Once these major revisions are implemented, the manuscript can be reconsidered for publication.

Reviewer 1 ·

Basic reporting

Dear Editor and Authors,
The review manuscript is very well written, and addresses a current and relevant topic for plant research.
From my point of view (not a native English speaker) the English is clear, unambiguous and correct throughout the text.
The literature used is sufficient to support the content. However, although I know that the topic is quite recent, I would like to see more articles from 2024 and even 2025 being cited in the text.
In general, the structure of the manuscript is in accordance with the requirements of the journal.
Please note the use of capital letters in titles and subtitles, and the need to create the summarized topics of Methodology, Results, and Conclusion before the abstract.
The review is of broad and interdisciplinary interest and is within the scope of the journal. There are few review studies in this area.
The introduction adequately presents the subject and highlights the audience and motivation for the study.

Experimental design

The content of the article is consistent with the aims and scope of the journal.
It appears that a rigorous investigation was conducted.
The methods were described in sufficient detail and information to replicate.
The research methodology is consistent.
The sources were cited appropriately, except for line 303 (Gu et al.) where the year is missing.
The review was organized logically.

Validity of the findings

The conclusion and outlook are in accordance with the objective of the study.

Additional comments

Suggestions
In lines 116, 228 and 267 use tsRNAs instead of TSRNA
The manuscript would be better if it presented some new results. For example, some in silico analysis from the tsRNA database, which the authors themselves cite. Some heatmaps?

Reviewer 2 ·

Basic reporting

The authors presented a review on the roles of tsRNAs in stress response in plants. Where the novelty, biological roles, and molecular mechanisms of tsRNAs have been described in details in the review. The recent advancement in the classification and biogenesis of tsRNAs was evaluated and summarized. The field has mostly been reviewed in human and lesser in plants, therefore the review is only general interest.
However, even if the article is structured, there are a few points that require more clarification.

The introduction, although covered the basic information on tsRNAs, requires more content and references, as of now, the section feels short of adequate information:
Line 37-38: "Finally, after various post-transcriptional modifications.." It would be ideal to expand on the term "various" and give a few examples. The usage of various is ambiguous, without specifying what post-transcriptional modifications, because that could mean any modifications.
Line 39-41: "In response to environmental stress.." What signaling pathways and physiological changes do plants undergo to respond to environmental stresses? again the usage of words are ambiguous without giving at least one example of pathways and physiological changes.
Line 41-42: "MicroRNAs (miRNAs) are important non-coding RNAs .." Without specifying which miRNAs and what type of stress, it might lead the readers to assume that all miRNAs are involved in all stress response in all plant species.
Line 43-44: "tRNAs generates tsRNAs, which influence cellular mechanisms similarly to miRNAs..." It would be beneficial to give a short description to how miRNAs influence cellular mechanisms, because this statement currently is assuming that readers are familiar with how miRNAs influence cellular mechanisms and referencing to something that have not been discussed.
Line 45: "especially under stress or disease conditions.." It needs elaboration on the stress and disease conditions, since the review is focused on plants, it should specify that.
Line 49: "PIWI-interacting RNAs (piRNAs) .." first time introducing piRNAs and without explaining what they are and what they do instead just to immediately followed with the statement "can be identified through small RNA sequencing". Needs reconsidering.
Line 56: "abiotic stress .." would be interesting to discuss why only abiotic stress is covered and not biotic, is it because there are limited studies?

Overall, the two bodies of the introduction felt disconnected, and lacking details that will give the readers a broader view of the topic.

Experimental design

The authors should consider restructuring the review, the current structure made the review hard to follow. There are comments within the main body of the review that the authors should consider addressing. Detail comments are below:

Classification and biogenesis of tsRNAs
Line 84-88:"...... they are produced by the cleavage of the anticodon loop in mature rRNAs by ribonuclease A (RNase A) or ribonuclease T2 (RNase T2). Could be interesting to talk about under what conditions are RNase A utilized compared to RNase, and vice versa.
Line 88-90 "The biogenesis of tiRNAs..." The period after pathway is not necessary, it can be one continuous sentence. "The biogenesis of tiRNAs originates from a non-Dicer-dependent pathway, where Dicer belongs to the RNase III family of endoribonuclease ..."
Line 90-94 "In mammalian cells..." interesting that authors decided to focus on tiRNA in mammalian cells without also talking about plant cells. Is the information not present in literature?
Line 95 "tRNA-derived fragment (tRFs) are a type of tsRNAs..." I think the opening of line 82 should be rephrased to better tie-in with line 95, or at least state that tRFs are 1 of the 2 main types of tsRNAs.
Line 111-114: "to address the inconsistent naming..." Suggest adding the literatures that proposed different naming systems (Kumar et al., 2014, PMID: 25392422; Pliatsika et al., 2017, PMID: 29181503) to support the statement of "inconsistent naming".

Subcellular localization of tsRNAs biogenesis
Line 116: assuming it should be lower case ts?
this section feels fragmented, statements should be rearranged for better flow.
Line 121-122: "Ma, Liu & Cao, (2021) pointed out..." this statement could be moved to after "controversial" at Line 118. Just because two separate statements talking about RNase T2 are separated by this statement. This section mainly talks about the localization of the mediator family of the tsRNA biogenesis, a reference is needed for line 123-124 regarding cytoplasmic RNase T2 can generate many tsRNAs in the cytoplasm. Future elaboration is needed for line 124-126, what are the established function of miRNAs in the ER? It is assuming the readers possessed the knowledge of miRNA functions in the ER.

Biological functions of tsRNAs
Line 130-131: "tRFs and tiRNAs, as small non-coding RNAs (sncRNAs) ..." Introduced a new term that should be introduced earlier in the review.
Line 131-137: "For instance..." this sentence is unusually long, consider rephrasing. Should clarify "they can act as miRNAs", under what circumstances they can act as miRNAs? clarify "serve as paternal epigenetic factors...", citations?
Line 137-139: "tRFs exhibits ...", microRNAs instead miRNAs? not the first time introducing the term, suggest just use the abbreviation for consistency.
Line138-139: "some previously.." are there any examples of the previously described miRNAs that might be tRFs? Would be beneficially to talk about how was this conclusion made.
It is interesting that authors first talked about tRFs and tiRNAs can act as miRNAs, then proceeded to talk about tRFs exhibiting functional similarities to miRNAs again. Should consider rephrasing to ensure a smoother flow.

tsRNAs suppress transcription
Line 144-145: "Notably..." when referring to tsRNAs here, does it referring to tiRNA? tRFs? or both. The reference experiment appeared to be using tRFs only. tRFs-PIWI in mammalian cells appeared to suppress transcription, but are there replication of this observation in plants? If not, it needs to be specified that this was observed in animals.

tsRNA regulate mRNA stability
Line 154: "Kumar et al., 2014 found that tsRNAs..." Would be ideal to know if the authors referring to tsRNAs as a whole or tRFs only, since the reference talks only about tRFs and the whole section appeared to be tRFs specific.
Line 171-173: "2-tRFs .." this is reinstating the statement from previous section.

tsRNAs regulate translation
Line 182: "IRES..." first time introducing an abbreviation without giving the full name, although functions followed, but readers still have no idea what it is.

tsRNAs regulate ribosome biogenesis

Line 212-214: "There is no direct evidence..." It is very confusing in terms of the context of these two sentences. The last statement regarding translation is not under the subsection that talks about specifically about translation.

The biggest problem is the confusion in subsection headers, many of those sections are tRFs specific, yet the term tsRNAs are used repeatedly suggesting the involvement of tiRNAs. The authors should specify the involvement of tiRNAs and tRFs independently like what has been done in line 212-213, or consider renaming the subsection headers.
Just a general comment, that it is interesting that more plant studies are tRFs based, while the majority of tiRNAs are studied in animal models.

Regulation of plant gene expression by tsRNAs in response to abiotic stress

Line 228: tsRNAs instead of TSRNAS
Line 229-230: "although..." suggesting tsRNAs accumulations are stress-specific?
Line 236: "the lower abundance.." relative low abundance? Otherwise, lower abundance in the mutant might only imply a directional difference but does not specify the difference is relative or significant. Could be misleading.
Line 237-240: might consider Luan et al., 2017 PMID 32635887 on exogenous ABA application. In addition, the period after Alves et al., 2017 is not necessary, it can be one continuous sentence.
Line 241-245: "in addition.." this talked about formation/biosynthesis of tsRNAs, but should clarify
how is this related to stress response?
Line 248-252: "in the context .." should elaborate on how transposon suppression is relevant to stress response.
Line 260-261: "from functional tRNAs that can enhance translation .." Ambiguous statement, should clarify.

Research methods of tsRNAs in plants
Line 267: tsRNAs instead of TSRNAS

Validity of the findings

While the review covered the topics in great details, the review suffers organization issues, I highly suggest that the authors format subsections in the main body following the same orders in the conclusion.
Majority of the unresolved questions/gaps are discussed in good details in the conclusion, however, future perspectives lack the same details. For example, line 360-361 "these questions require further investigation" without proposing methods or experiments to how they should be carry out.

Reviewer 3 ·

Basic reporting

Clear and unambiguous, professional English used throughout

-->yes

Literature references, sufficient field background/context provided.

--> yes

Professional article structure, figures, tables. Raw data shared.
--> Yes, ta white background would be better for the figures compared to the skyblue background

Is the review of broad and cross-disciplinary interest and within the scope of the journal?
--> for broad interest but not cross-disciplinary, within the scope of the journal

Has the field been reviewed recently? If so, is there a good reason for this review (different point of view, accessible to a different audience, etc.)?
--> yes

Does the Introduction adequately introduce the subject and make it clear who the audience is/what the motivation is?
--> I think it is. An introduction to small RNA machinery along with figure would help the readers to catch more easily

Experimental design

the design looks ok

for the abiotic stress section, subsections could be introduced.

Validity of the findings

Impact and novelty not assessed. Meaningful replication encouraged where rationale & benefit to literature is clearly stated.
--> Not many reviews are there on this topic

Conclusions are well stated, linked to original research question & limited to supporting results.
-->Conclusion is too long, this can can be moved to different section topics

Is there a well developed and supported argument that meets the goals set out in the Introduction?
Does the Conclusion identify unresolved questions / gaps / future directions?

--> not fully. to some extent

---

## Round 0.2 · accepted · Accept

Congratulations on the acceptance of your manuscript.

Reviewer 2 ·

Basic reporting

I think after the thorough revision, the manuscript is suitable for publication.

Experimental design

no comment.

Validity of the findings

no comment.